# Position: Efficient General Intelligence requires Neuro-Symbolic Integration: Pillars, Benchmarks, and Beyond

## Abstract

Recent breakthroughs in Large Language Model (LLM) development have rekindled hopes for broadly capable artificial intelligence. Yet, these models still exhibit notable limitations – particularly in deductive reasoning and efficient skill acquisition. In contrast, *neuro-symbolic* approaches, which integrate sub-symbolic pattern extraction with explicit logical structures, offer more robust generalization across diverse tasks. We argue that additional factors – such as modular transparency, flexible representations, and targeted prior knowledge – are crucial to further enhance this generalization. Our analysis of both historical and contemporary AI methods suggests that **a multi-component neuro-symbolic implementation strategy is necessary for efficient general intelligence**. This position is reinforced by the latest performance gains on the ARC-AGI benchmark and by concrete case studies demonstrating how neuro-symbolic designs address gaps left by purely neural or purely symbolic systems.

## 1. Introduction

**We posit that truly general artificial intelligence demands the unification of data-driven neural approaches with the interpretability and compositional expressiveness of symbolic methods.** Across decades of progress, research on "artificial intelligence" has often centered on narrow tasks and small leaps in computational automation, without necessarily pursuing robust, human-like intelligence. This changed with the rise of large, monolithic neural networks – models that excel in pattern extraction and display intriguing emergent capacities (Bubeck et al., 2023). Yet, while these black-box approaches are remarkable in many respects, they also suffer from opaque decision-making processes and often exhibit only **local** forms of generalization. They thus provide limited insights into the core mechanisms underlying *flexible, human-level* intelligence.

Motivated by these gaps, an increasing number of researchers suggest incorporating symbolic reasoning or "transparent" inductive biases into deep learning pipelines, giving rise to *neuro-symbolic* approaches (d'Avila Garcez & Lamb, 2023; Keber et al., 2024). By preserving the neural model's strengths in statistical pattern recognition and combining them with symbolic structures that allow for compositional logic, explainable decisions, and interpretability, neuro-symbolic methods promise broader skill-acquisition efficiency, deeper semantic understanding, and safer real-world deployment (Hernández-Orallo, 2020; Hassija et al., 2024).

However, merely layering symbolic modules on top of neural back-ends does not automatically confer *general* intelligence. To foster meaningful progress, we must (i) clarify the very notion of intelligence, especially in terms of *skill acquisition* and *generalization difficulty*, and (ii) pinpoint how best to evaluate a model's capacity to abstract knowledge from sparse data and adapt to novel tasks. Below, we outline why Chollet (2019)'s emphasis on "skill-acquisition efficiency" is particularly fruitful, how debates on *behaviorism versus internalism* push us to seek more transparent model mechanisms, and why *benchmarks* specifically designed for general intelligence play such a central role. We then highlight the Abstraction and Reasoning Corpus (ARC) and its latest ARC-AGI-1 challenge (Chollet et al., 2025) as a tangible testbed for demonstrating how neuro-symbolic methods could open doors to true breadth of reasoning ability.

### 1.1. Defining Intelligence

Despite centuries of study, intelligence remains notoriously difficult to define comprehensively (Legg & Hutter, 2007). We adopt the formulation by Chollet (2019) that views the intelligence of a system as "a measure of its skill-acquisition efficiency over a scope of tasks, with respect to priors, experience, and generalization difficulty." This perspective shifts attention from raw *performance* on a single task to the *ability to learn new* tasks under constraints – such as limited data, novel transformations, or minimal prior knowledge. Indeed, skill-acquisition efficiency is at the heart of what sets "general" intelligence apart from specialized or over-engineered solutions (Bober-Irizar & Banerjee, 2024).

## 1.2. Behaviorism vs. Internalism

A longstanding philosophical debate pertains to whether only *external* behavior matters (behaviorism) or whether the *internal* mechanisms of thought carry essential explanatory value (internalism). In contemporary machine learning, this tension appears as "functionality vs. interpretability" or "black-box vs. transparent systems." High-performing but opaque models – like many Large Language Models – demonstrate that achieving sophisticated outputs does not necessarily illuminate the process by which the model reasons (Hernández-Orallo, 2020; Schlangen, 2021).

As these systems are deployed in sensitive or high-stakes environments, interpretability and control become paramount (Hassija et al., 2024). Post-hoc explanations often provide only a partial window into massive parameter spaces, leaving significant uncertainties about *why* a particular decision was reached (Kenny et al., 2021; Slack et al., 2021; Leemann et al., 2023; Rong et al., 2023). By contrast, **inherent model transparency** – via symbolic modules, meaningful structured interfaces, or modular architectures – can yield more reliable comprehension of internal processes, facilitate debugging, and bolster trustworthiness. Consequently, we argue that *internalist* considerations should shape the development of any model that aspires to broader, more systematic intelligence.

## 1.3. Generalization Efficiency

Even when a model attains notable performance on a suite of tasks, it is crucial to distinguish between *intrinsic generalization* and *engineered solutions*. Many recent successes hinge on massive data curation, architectural tuning, or manual injection of priors – leading to impressive system-centric results, but not necessarily reflecting a model's capacity to *autonomously* learn how to solve unseen tasks. As Chollet (2019) notes, a "developer-aware" perspective on skill acquisition controls for these extra-human interventions. Without such a perspective, higher benchmark scores risk being misread as general intelligence.

Hence, if the field's ambition is true *general* intelligence – rather than a proliferation of specialized or heavily handcrafted solutions – then adopting metrics and methods highlighting *skill-acquisition efficiency* becomes indispensable. This, in turn, requires reliable ways to evaluate how well a model performs under low-data, unseen, or compositional scenarios – where brute-force training or naive memorization is infeasible.

## 1.4. Benchmarking for Generality

The search for a benchmark that isolates genuine abstraction and reasoning from mere pattern fitting has led to the Abstraction and Reasoning Corpus (ARC) (Chollet, 2019),

later extended into ARC-AGI-1 (Chollet et al., 2025). Unlike tasks saturated by large, curated datasets, ARC consists of small, diverse puzzles that test "core knowledge" concepts like spatial manipulation, color/object transformations, or compositional logic (Moskvichev et al., 2023).

Despite being straightforward for humans, ARC tasks have proven unexpectedly difficult for computational models, with only about half the tasks consistently solved (Bober-Irizar & Banerjee, 2024; ARC Prize, 2024). This difficulty emerges precisely because ARC demands *abstract generalization* over a minimal set of examples, thwarting superficial shortcuts. While ARC alone is not a perfect proxy for all human-level reasoning (Chollet, 2019), it remains a valuable gauge of small-data adaptability, creative knowledge transfer, and flexible problem solving. Table 1 summarizes the key dimensions for designing models with broad generalization capabilities.

In what follows, we leverage ARC-AGI-1 to motivate why **hybrid neuro-symbolic architectures** – infused with explicit mechanisms for transparency, compositional reasoning, and high-level knowledge abstraction – are integral to bridging the gap from narrow task competence to more truly *general* intelligence.

# 2. Alternative Views

In this position paper, we argue that *modular neuro-symbolic integration* is central for achieving efficient generalization. Nevertheless, it is important to recognize other significant perspectives in the field, especially since some researchers propose that *either* purely neural or purely symbolic methods might suffice if combined with enough data, computational resources, or engineering effort. Below, we discuss these alternatives – large language models (LLMs) as an archetype of purely neural approaches, and domain-specific languages (DSLs) or program-synthesis approaches as a representative of purely symbolic strategies – and evaluate why they each have notable strengths yet ultimately fall short when it comes to *broadly* efficient generalization.

## 2.1. Purely Neural Approaches: Large Language Models

Transformers and large language models (LLMs) have undeniably exhibited broad emergent capabilities, including surprising generalization and few-shot reasoning, across multiple domains (Bubeck et al., 2023; Webb et al., 2023). Remarkably, they can perform competitively even on the Abstraction and Reasoning Corpus (ARC) when equipped with skillful prompting, chain-of-thought techniques, and *self-improvement* querying (Greenblatt, 2024; Berman, 2024; ARC Prize, 2025; Chollet et al., 2025). Indeed, GPT-4, Sonnet 3.5, and *o3* consistently achieve the highest ARC-AGI-1

*Table 1.* Key Dimensions for Designing Models with Broad Generalization

| Dimension | Importance for General Intelligence | Representative Works |
|---|---|---|
| **Skill-Acquisition Efficiency** | Emphasizes how well a system adapts to new tasks without extensive retraining; penalizes overreliance on developer engineering or huge datasets. | (Chollet, 2019), (Bober-Irizar & Banerjee, 2024) |
| **Transparency & Interpretability** | Strengthens trust and debugging; post-hoc explanations are often insufficient for large black-box models. Inherent transparency is crucial for real-world reliability. | (Hernández-Orallo, 2020), (Hassija et al., 2024) |
| **Symbolic Reasoning** | Allows compositional, logically coherent transformations. Fosters human-level abstraction and provides robust handling of discrete structures. | (d'Avila Garcez & Lamb, 2023), (Keber et al., 2024) |
| **Neural Representations** | Harnesses powerful pattern-extraction capabilities from raw data (images, text), enabling feature discovery and capturing nuanced correlations. | (Bubeck et al., 2023) |
| **Small-Data Adaptation** | Avoids brute-forcing solutions by demanding strong generalization from very few examples (as in ARC tasks), exposing true abstraction capabilities. | (Moskvichev et al., 2023), (Chollet et al., 2025) |

public scores when allowed large-scale test-time optimization. This level of success leads many researchers to view LLMs as the foundation for future general-purpose AI.

**Strengths of LLMs.** Modern LLMs have several strengths ranging from a wide knowledge coverage to reasoning capabilities and flexibility when applied to downstream tasks.

- **Pre-training on Massive Corpora** allows for extensive self-supervised learning on diverse text sources. In this way, LLMs acquire a wealth of representations, effectively consolidating and covering wide-ranging knowledge (Bubeck et al., 2023).

- **Flexible Transfer of Knowledge** can be applied to handle various downstream tasks (including non-linguistic tasks expressed in language) with minimal fine-tuning, thanks to in-context learning capabilities and powerful embedding spaces (Dong et al., 2023; Berman, 2024).

- **Emergent Reasoning Behaviors** can be elicited through prompting strategies such as chain-of-thought or retrieval augmented generation. Such reasoning-like procedures within LLMs often improve the performance on complex tasks (Webb et al., 2023).

**Challenges and Limitations.** Despite impressive benchmarks, purely neural methods still exhibit significant hurdles regarding *efficient* generalization:

1. **Opaque and Brittle Emergence:** The extent to which LLMs can perform genuine abstract reasoning (versus pattern matching) remains an open debate (Valmeekam et al., 2023; Kaddour et al., 2023; Dziri et al., 2023; Lewis & Mitchell, 2024; Wang et al., 2024; Lotfi et al., 2024; Schuurmans et al., 2024). Their "emergent" abilities can be unreliable, hard to interpret, and domain-specific (Bober-Irizar & Banerjee, 2024).

2. **Data-Hungry and Costly:** Training large-scale transformers demands massive, human-generated corpora – and some fear we are reaching the upper limit of high-quality data for further scaling this approach (Sutskever, 2024). In addition, fine-tuning or test-time brute forcing can be expensive and inefficient (Sachdeva et al., 2024).

3. **Developer vs. Model Intelligence:** Many LLM-based successes rely heavily on *engineered prompting* and human-coded heuristics. Thus, high-level performance may reflect *developer-centric* skill more than an intrinsic model capacity for generalization (Chollet, 2019; Dong et al., 2023; Yu et al., 2023; Bober-Irizar & Banerjee, 2024).

4. **Lack of Transparency:** Unlike modular designs, LLMs encode reasoning steps in vast weight matrices, limiting interpretability. This black-box nature impedes deeper analysis of the reasoning process and complicates improvements targeted at genuine compositional intelligence (Garcez & Lamb, 2023).

Moreover, recent ARC results reveal that while LLM-based approaches can outperform other methods on the public

benchmark, they do so through *massive prompt engineering* or resource-intensive test-time synthesis (Greenblatt, 2024; Berman, 2024). Mahowald et al. (2024) draw parallels to the human brain's specialized "language areas," cautioning that forcing a language-dominant model to cover abstract non-linguistic tasks may be fundamentally inefficient. Hence, even though LLMs are powerful in practice, they are less suitable as an *academic research framework* for understanding the *mechanisms* behind generalization.

**Conclusion for LLMs.** While purely neural approaches have reshaped modern AI, purely monolithic LLMs appear suboptimal as a basis for *broad* efficiency. Large data combined with sufficient computing resources can brute force solutions, but they do not illuminate the core processes underlying abstract reasoning. For those interested in deeper interpretability, explainability, or developer-aware skill acquisition, *neuro-symbolic integration* seems indispensable.

### 2.2. Purely Symbolic Approaches: Domain-Specific Languages and Program Synthesis

Although overshadowed by neural methods in recent years, purely symbolic or logic-based AI once dominated the field and retains a devoted following (Kastner & Hong, 1984). Within the ARC domain, the most visible symbolic attempts revolve around exhaustive search in a *Domain-Specific Language* (DSL) or program-synthesis methods such as Dream-Coder (Ellis et al., 2020).

**DSL-Based Methods.** Early top-ranked solutions in the original ARC challenge relied on large, hand-crafted DSLs (icecuber, 2020; de Miquel, 2020; Larchenko, 2020). By systematically searching over a predefined set of transformations and heuristics, these approaches found valid transformations for specific puzzles. However, these DSL-based methods achieved only modest coverage due to the combinatorial explosion of possible transformations and the diversity of ARC tasks. They also demanded extensive human engineering to hard-code each concept, undermining *developer-aware* generalization measures (Bober-Irizar & Banerjee, 2024).

**Program Synthesis Approaches.** Program-synthesis frameworks like DreamCoder (Ellis et al., 2020) extend the DSL idea with higher-level constructs (e.g., control-flow operators, recursion). While this unlocks greater expressiveness, it can also inflate the search space. Adapting a fully general programming language for ARC tasks becomes cumbersome because ARC-AGI-1 is already quite challenging without further increasing the solution space (Bober-Irizar & Banerjee, 2024).

**Symbolic Drawbacks.** While symbolic approaches can offer strong interpretability (one can often track each logical step explicitly), they typically struggle to infer abstract "core concepts" from limited data without some learned inductive biases. Their purely top-down logic has trouble coping with the noisy, high-dimensional input distributions where data-driven feature extraction is crucial. Additionally, naive symbolic search tends to be fragile in the face of tasks requiring approximate or probabilistic reasoning.

**Conclusion for Symbolic Methods.** Historically, purely symbolic solutions have rarely scaled well across diverse tasks and have difficulty encoding robust priors for low-data settings (Kastner & Hong, 1984; Ellis et al., 2020). The *ARC experience* confirms that exhaustive or highly engineered symbolic DSLs rapidly reach diminishing returns. Hence, purely symbolic approaches, while valuable for interpretability and logic, alone are insufficient for broad or efficient generalization.

### 2.3. Synthesis of Both Views

In summary, purely neural approaches (e.g., LLMs) can demonstrate remarkable capabilities but often rely on extensive engineering, computational resources, and data, with limited inherent interpretability. Purely symbolic approaches retain logical clarity but cannot cope effectively with the complexity and ambiguity that general tasks demand. We, therefore, see *neuro-symbolic integration* – the systematic coupling of learnable neural components with explicit symbolic representations and reasoning – as the most promising route to truly efficient, transparent generalization, far beyond either paradigm alone.

## 3. Neuro-Symbolic Approaches

Neuro-symbolic methods stand at the intersection of statistical learning and explicit symbolic reasoning, offering a promising path toward *efficient* generalization. As discussed in Section 2, purely neural or purely symbolic methods each have strengths, but neither alone excels at developer-aware skill acquisition. Nevertheless, these two paradigms clearly complement each other (Bober-Irizar & Banerjee, 2024), already hinting at the power of *neuro-symbolic integration* to tackle a broader range of tasks (Bober-Irizar & Banerjee, 2024; Chollet et al., 2025).

Multiple works have surveyed the general advantages and disadvantages of neuro-symbolic approaches in depth (Hamilton et al., 2022; Hitzler et al., 2022; Garcez & Lamb, 2023; Keber et al., 2024; Bhuyan et al., 2024). Rather than revisiting all of these aspects, we focus here on the key *generalization* benefits, underscored by the most recent ARC-AGI-1 findings (Chollet et al., 2025).

**Terminology.** The term "neuro-symbolic" (sometimes abbreviated "NeSy") can encompass a wide variety of hybrid architectures and learning strategies. While the specific mechanisms vary, the core idea is to marry *symbolic structures* (e.g., logic programs, DSLs, knowledge graphs) with *neural components* (e.g., deep networks or learned embeddings) (Hitzler et al., 2022; Garcez & Lamb, 2023; Keber et al., 2024).

Table 2 summarizes the main aspects of representative state-of-the-art NeSy approaches for generalization in ARC-like tasks.

**Coming from the Neural Side.** In Large Language Models (LLMs), one could argue that chain-of-thought prompting or structured "reasoning graphs" already hint at neuro-symbolic principles (Hitzler et al., 2022; Keber et al., 2024). These techniques often wrap a neural transformer in a scaffold of symbolic instructions or constraints (Yu et al., 2023), thus improving performance across multiple tasks. For example, Xu et al. (2024) demonstrate how extensive logical orchestration around LLM calls boosts reliability on diverse tasks. Notably, the top LLM-based ARC-AGI-1 approaches also incorporate symbolic heuristics to stabilize generalization – further evidence that *purely* sub-symbolic solutions remain insufficient (Franzen et al., 2024; Barbadillo, 2024; Chollet et al., 2025).

**Coming from the Symbolic Side.** Conversely, the golden era of symbolic AI faded in the late 1980s, giving way to sub-symbolic (neural) approaches. However, the limitations that once suffocated symbolic AI – such as brittle rule systems or exponential search complexity – can be mitigated by modern neural advances and computing power (Mira, 2008). Rather than reviving purely symbolic methods, researchers increasingly aim to harness the strengths of both paradigms: explicit logic for interpretability and systematic abstraction, and neural modules for data-driven feature extraction and robustness (Hitzler et al., 2022; Garcez & Lamb, 2023). A clear illustration is Bober-Irizar & Banerjee (2024), who build upon a DSL-based ARC solver by adding learnable "concept formation" components, significantly boosting efficiency and success rates.

**Synergy in Practice.** One prominent driver behind neuro-symbolic integration is *generalization efficiency* (Bhuyan et al., 2024). Hybrid models can learn abstract concepts more compactly, leveraging both (i) a neural module to handle noisy or high-dimensional inputs and (ii) a symbolic module to enforce logical coherence and compositional reasoning. This synergy is particularly relevant in low-data tasks like ARC, where purely neural systems often overfit, and purely symbolic systems lack robust inductive priors. While recent work has demonstrated promising gains on ARC (Moskvichev et al., 2023; Chollet et al., 2025; Bober-Irizar & Banerjee, 2024), open challenges remain – most notably:

- **Exploding Search Spaces.** Combining symbolic search with neural heuristics can mitigate the worst-case combinatorial complexity explosion, but designing these heuristics remains nontrivial (Bober-Irizar & Banerjee, 2024).

- **Data Efficiency vs. Model Complexity.** ARC-AGI-1 tasks demand strong reasoning from minimal examples, stressing the importance of balanced architectures that do not over-parameterize (Moskvichev et al., 2023).

- **Formation of New Concepts.** Handling ever-evolving domains requires neuro-symbolic methods that can *learn new concepts dynamically* rather than rely solely on a hard-coded DSL (Bober-Irizar & Banerjee, 2024).

Though these obstacles are significant, the ability of neuro-symbolic methods to unify inductive and deductive reasoning is an especially potent strength – analogous to "System 1" vs. "System 2" thinking in human cognition (Kahneman, 2011; Garcez & Lamb, 2023). As computational and data constraints grow more urgent, *this marriage of neural and symbolic approaches* will likely become not just beneficial but *indispensable*.

**Closing the Gap.** Ultimately, the goal of neuro-symbolic research is to exploit each paradigm's complementary strengths: neural networks excel at fast, intuitive processing of raw data, whereas symbolic formalisms enable explicit logic and compositional abstraction. By weaving these together, a system can move beyond behavioristic success into genuine *skill-acquisition efficiency*, operating effectively with minimal data or developer engineering while staying transparent, interpretable, and controllable. In the following sections, we delve deeper into the specific design components and synergy effects that make neuro-symbolic architectures uniquely suited to achieving broader generalization.

## 4. Pillars of Efficient Neuro-Symbolic Generalization

Achieving robust generalization through neuro-symbolic methods requires more than simply pairing a neural module with a symbolic one. As Odense & Garcez (2022)(p. 38) argue, the key is to exploit the "complementary strengths and weaknesses" of both connectionist and symbolic paradigms – rather than letting one approximate or overshadow the other. Consequently, the central question in the years ahead is *how* to fuse these components so that they collectively yield effective skill acquisition across diverse tasks. We propose

*Table 2.* Representative Neuro-Symbolic Approaches for Generalization in ARC-like Tasks

| Approach | Neural Component | Symbolic Component | Key Mechanism & Insights |
|---|---|---|---|
| **Bober-Irizar & Banerjee (2024)** (Bober-Irizar & Banerjee, 2024) | Learned concept-formation module (e.g., CNN-like embeddings to identify object features) | DSL-based program search for transformations | Uses neural heuristics to guide symbolic search, significantly reducing the DSL's combinatorial explosion. Demonstrates notable gains on ARC tasks versus purely symbolic baselines. |
| **SearChain** (Xu et al., 2024) | Large Language Model (transformer) for reasoning over prompts | Search framework with symbolic constraints (e.g., BFS or rule-based expansions) | Combines "search in the chain" logic with LLM prompting; symbolic scaffolding constrains the neural model's proposed transformations, improving reliability on diverse puzzle-solving tasks. |
| **DreamCoder** (Ellis et al., 2020) | Neural "wake-sleep" cycle that learns common subroutines or concepts | Inductive program synthesis in a high-level language (with control-flow, recursion) | Iteratively refines a library of reusable functions—symbolic *abstractions*—guided by neural scoring. *DreamCoder* is not specifically designed for ARC but illustrates how learned domain knowledge can be symbolically encoded. |
| **Neuro-Symbolic DSL Enhancements (various)** (Hamilton et al., 2022; Hitzler et al., 2022; Garcez & Lamb, 2023; Bhuyan et al., 2024) | Neural embeddings for object detection, classification, or spatial feature extraction | Logic-based DSL or ontology enforcing compositional rules | General family of hybrid methods: neural modules handle perceptual tasks or fuzzy matches, while symbolic DSL enforces interpretability and constraint satisfaction. Shown to improve data-efficiency and interpretability on small "grid-world" or ARC-like puzzles. |

that the following **fundamental pillars** are indispensable for attaining *efficient* (developer-aware) generalization in neuro-symbolic systems.

### 4.1. Multi-Component Synergy Effects

Although merging neural and symbolic layers is crucial, other "side problems" – such as representation strategies, uncertainty handling, and knowledge encoding – are equally significant for broad-scope generalization (Bhuyan et al., 2024). Many advanced methods overlook at least one dimension (e.g. using trivial transformations or underpowered representations), losing potential flexibility (Franzen et al., 2024; Berman, 2024). In contrast, a *systematic* approach that addresses each sub-component fosters powerful synergy effects between them(Garcez & Lamb, 2023).

While modular integration entails substantial engineering (Garcez & Lamb, 2023) and necessitates careful data-format alignment, Bober-Irizar & Banerjee (2024) demonstrate its worth: their neuro-symbolic concept-formation technique, inspired by Ellis et al. (2020), overshadow naive DSL search by leveraging richer learned representations and heuristics to prune the search space.

**Takeaway**: By *holistically* optimizing each component in the system, one transcends individual contributions and achieves system-wide synergy, enabling more capable and efficient generalization.

### 4.2. Model Specificity

Global ambitions need not result in "solve-all" monstrosities. Instead, concentrating on a well-defined domain – such as ARC's core priors – prevents runaway complexity (Chollet et al., 2025). Despite ARC's seemingly simple puzzles (e.g. shape manipulation), thorough mastery proves nontrivial without the "core knowledge" priors they were constructed from (Chollet, 2019; Ellis et al., 2020). Meanwhile, large foundation models often require careful prompting to isolate relevant priors (Greenblatt, 2024; Berman, 2024). Their strengths in specialized world knowledge or linguistic reasoning are not relevant for ARC(Chollet et al., 2025)(p.1).

**Takeaway**: A *targeted model scope*, with sufficient coverage of relevant key primitives yet focused capabilities, yields a broad solution space while still being feasible and tractable.

*Table 3.* Key Pillars of Efficient Neuro-Symbolic Generalization

| Pillar | Why It Matters | Key Challenges | Representative Works |
|---|---|---|---|
| **Multi-Component Synergy (4.1)** | Achieves more powerful system-wide effects transcending capacities of individual components | Engineering complexity; aligning data formats and abstraction levels of modules (e.g. integrating neural (pattern extraction) and symbolic (logical composition) capacities for broader coverage) | (Ellis et al., 2020; Garcez & Lamb, 2023; Bober-Irizar & Banerjee, 2024) |
| **Model Specificity (4.2)** | Prevents scope bloat by focusing on well-defined core concepts; avoids extraneous features | Balancing breadth vs. tractability; ensuring fundamental priors are covered | (Chollet, 2019; Chollet et al., 2025) |
| **Knowledge Encoding (4.3)** | Embeds abstract human insights, reducing the burden of brute-force or data-heavy engineering | Over-encoding task-specific solutions; selecting which concepts to "hard-code" vs. learn | (icecuber, 2020; de Miquel, 2020; Bober-Irizar & Banerjee, 2024) |
| **Knowledge Acquisition & Transfer (4.4)** | Captures new concepts from training data and reuses them adaptively at test time | Designing effective but flexible training paradigms (e.g. curriculum learning, test-time fine-tuning) | (Ellis et al., 2020; Akyürek et al., 2024; Bober-Irizar & Banerjee, 2024; Chollet et al., 2025) |
| **Representation (4.5)** | Governs how data is parsed and manipulated (object-based, graph-based, etc.), greatly influencing efficiency and model scope | Deciding the optimal abstraction level (e.g., pixels vs. objects) and bridging neural embeddings with symbolic structures | (Xu et al., 2023a; Skean et al., 2024; Barbadillo, 2024) |
| **Abstractions & Hierarchies (4.6)** | Filters superfluous detail, enabling compositional reasoning and meaningful representations | Choosing the number and granularity of layers; ensuring each abstraction captures meaningful transformations | (Krizhevsky et al., 2017; Xu et al., 2023b) |

### 4.3. Knowledge Encoding

Symbolic frameworks excel at instilling human knowledge, yet enumerating solution scripts for each task kills adaptability. Instead, defining *process-level* abstractions (e.g. "move(object, vector)") fosters reusability across countless tasks (Xu et al., 2023a). Ellis et al. (2020, p. 18) emphasize that "rich systems of built-in knowledge" radically accelerate learning – a stance aligning with the principle that *broad* competence arises from fundamental, composable operators. **Takeaway**: Injecting *abstract human expertise* (concept-level rather than solution-level) boosts data efficiency and encourages flexible reuse.

### 4.4. Knowledge Acquisition, Transfer, and Combination

No matter how thorough the initial knowledge encoding, new tasks inevitably appear. Thus, a neuro-symbolic system must *learn* fresh concepts during training and *recombine* them spontaneously at inference (Chollet, 2019). ARC-AGI-1 showcases how test-time fine-tuning (TTFT) can be essential for unseen tasks (Akyürek et al., 2024; Chollet et al., 2025). Likewise, DreamCoder's "sleep-wake" cycle continuously refines a library of existing abstractions (Ellis et al., 2020), which Bober-Irizar & Banerjee (2024) adapt to handle ARC's diverse puzzle types.

**Takeaway**: Flexible generalization arises from *continual concept formation* plus *dynamic adaptation* at test time.

### 4.5. Representation

Representational design profoundly shapes a system's ability to generalize. While neural embeddings capture latent structure, they can be overly broad for specialized tasks like ARC (Garcez & Lamb, 2023; Skean et al., 2024). On the

other hand, graph- or object-centric representations simplify transformations (Xu et al., 2023a), thus reducing search complexity and clarifying model behavior. Replacing pixel-level manipulations with object-level reasoning, for instance, can diminish the needed symbolic operator set by an order of magnitude as respective ARC puzzles are situated on this abstraction level(Xu et al., 2023a). In contrast to abstraction capabilities, which are more processing-focused, representation spaces reflect the model's perspectives on the world (i.e., world model)(Huh et al., 2024; Barbadillo, 2024).

**Takeaway**: Accurately aligning representations with the *natural granularity* of the domain maintains interpretability and computational efficiency while setting a meaningful scope for the model.

### 4.6. Abstractions and Hierarchies

Layered abstractions are foundational to both human cognition and deep-network architectures (LeCun et al., 1989; Riesenhuber & Poggio, 1999; Grill-Spector & Malach, 2004; Krizhevsky et al., 2017). In the ARC-AGI-1 context, moving from pixel-level to object- or pattern-level operations delivers major efficiency improvements (Xu et al., 2023a;b). Each abstracted layer or module discards noisy details, accentuating shared structures across tasks while bolstering interpretability.

**Takeaway**: *Hierarchical design* combines low-level perception and high-level logic, enabling compositional reasoning and meaningful explanations/representations.

### 4.7. Concluding Remarks on the Pillars

Collectively, the six pillars in Table 3 – **multi-component synergy, model specificity, knowledge encoding, knowledge acquisition/transfer, representation, and hierarchical abstractions** – constitute the blueprint for efficient neuro-symbolic generalization. Of course, they are not exclusive to neuro-symbolic approaches and are potentially useful in other domains, too. When each is addressed deliberately and woven together cohesively, the *connectionist–symbolic* merger achieves far more than either paradigm alone. From DSL-based solutions fortified by learned heuristics (Bober-Irizar & Banerjee, 2024) to LLM-driven systems guided by symbolic constraints (ARC Prize, 2025), such interplay has already advanced complex reasoning tasks in ARC. We posit, therefore, that *fully engaging these pillars is indispensable* for the next leap in developer-aware, data-efficient, and transparently interpretable general AI.

## 5. Conclusion

The pillars outlined in Section 4 – from multi-component synergy and model specificity to hierarchical abstraction – reinforce and depend upon one another. Their interplay is precisely what enables systems to generalize, adapt, and recombine knowledge in new settings with minimal data or developer engineering. We contend that *this synergy* lies at the heart of intelligence itself: carefully crafted representations, thoughtfully curated curricula, and dynamic transfer strategies collectively drive skill-acquisition efficiency.

Interestingly, such modular-yet-intertwined specialization echoes the structure of the human brain. While AI need not mimic biology directly, the brain's functional organization offers strong evidence that partitioning cognition into co-operating modules – symbolic and neural – is both more efficient and more transparent than a monolithic design (Kahneman, 2011). The Abstraction and Reasoning Corpus (ARC) challenges highlight the difficulties and the *promise* of this approach (Chollet et al., 2025).

Indeed, *proper* neuro-symbolic integration has been recognized as crucial yet technically daunting (Garcez & Lamb, 2023; Chollet et al., 2025). Initial development costs may exceed those of monolithic solutions, but the reward – developer-aware, data-efficient systems that can flexibly adapt – is immense. Although our pillars are not fundamentally new in isolation, acknowledging how they interconnect offers a fresh lens on the essence of generalization.

To move the field forward, we propose **four priority directions**:

- **Benchmarks for Synergy**: More comprehensive test suites (i.e., by extending ARC-AGI-style tasks) to systematically measure how effective the particular strengths of symbolic and neural components are utilized.

- **Open-Source DSL-Neural Frameworks**: Facilitating modular experimentation to ensure that promising ideas can be tested swiftly and reproducibly.

- **Interpretable Module Integration**: Standardized protocols to visualize how learned representations and symbolic rules interact.

- **Safety and Trust**: Deploying neuro-symbolic designs in safety-critical domains (medical, autonomous systems) to enhance transparency and reliability.

We encourage the research community to intensify its focus on these *synergy-driven* neuro-symbolic methods. While achieving truly general intelligence is undeniably challenging, *starting now* and grappling with multi-component integration – rather than evading it – stands to unlock the next great leap in flexible, safe, and transparently interpretable general AI.

## Impact Statement

By advocating a synergy of data-driven and symbolic approaches, this work paves the way for AI systems that learn efficiently, reason transparently, and adapt efficiently to new challenges. While our primary goal is to advance machine learning methodologies, the ripple effects of more robust and interpretable AI could reshape various sectors – from healthcare and finance to education – by bolstering reliability and trustworthiness.

Compared to black-box models, neuro-symbolic solutions are inherently more amenable to auditing and control, mitigating risks of hidden biases or unintended behaviors. Such transparency is essential for aligning AI with societal values. We thus believe that emphasizing developer-aware intelligence, modular design, and clear human oversight offers a safer, more equitable foundation for the technology's continued evolution.

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
