# OpenReview forum: "Position: Efficient General Intelligence requires Neuro-Symbolic Integration: Pillars, Benchmarks, and Beyond"
_ICML.cc/2025/Position_Paper_Track — Submitted to ICML 2025 Position Paper Track_

### Official Review · Reviewer_tqCm · 2025-03-09

**Significance:** 3
**Argument Clarity:** 3
**Rating:** 2
**Confidence:** 4

**Questions:**

NA

**Discussion Potential:**

1

**Paper Summary:**

This paper advocates the position that achieving general intelligence requires incorporating neurosymbolic methods. The discussions include strengths and weaknesses of the pure neural methods and the neurosymbolic methods.

**Position:**

Yes

**Position In Title:**

Yes

**Related Work:**

2

**Strengths And Weaknesses:**

The position in this paper is a long-standing debate in the field and is of high relevance and importance to the ICML community. There are substantial numbers of researchers on both sides with strong convictions. Consequently, re-stating this topic is not novel. A good position paper on this topic needs to make meaningful contributions to the debate: maybe drawing people's attention to certain important results and making novel analysis, or maybe suggesting new experiments that will be revealing for the debate, or maybe suggesting new research directions, etc.

Unfortunately, the paper does not contribute enough to the debate. The stated strengths and weaknesses of the pure neural methods and the neurosymbolic methods would be agreed by most researchers. The arguments made for the position would be agreed by people who already believe in it, and there is not enough new information to change the mind of those who do not. I cannot find concrete new suggestions on experiments or research directions.

Another weakness of this paper is being overly reliant on ARC AGI as the benchmark.

**Support:**

3

---

> ### Author Rebuttal · Authors · 2025-03-31
>
> Thank you for your review. We appreciate your observations and would like to clarify the scope and contributions of our position paper.
>
> ## A) Clarifying Our Core Contribution
>
> ### A holistic approach beyond NeSy
> While we discuss neuro-symbolic (NeSy) methods, our goal is not to revisit the neural vs.\ symbolic debate but to present a broader, multi-component perspective on achieving efficient generalization.
>
> In **section 4**, we introduce a unified, six-component design perspective for researchers aiming at **efficient generalization**, whether through neuro-symbolic means or otherwise. These six pillars are orthogonal to classical NeSy design principles.
>
> Although broad, each pillar represents a distinct aspect of intelligent system development, from encoding knowledge and building abstractions to multi-component synergy. In point B with reviewer S6vw, we further clarify how these pillars differ and interact.
>
> We argue in section 4, that a narrow focus on neural-vs-symbolic trade-offs is insufficient for achieving human-like, data-efficient generalization. Both purely neural and purely symbolic approaches can attempt to incorporate some pillars, though often less effectively.
>
> ### Novelty
> Our work synthesizes recent empirical and theoretical findings, including results from January 2025 on the ARC challenge (GPT~o3, test-time fine-tuning, etc.).
> To our knowledge, no prior work has systematically mapped these six pillars or argued for their collective necessity in enabling true skill-acquisition efficiency.  While individual pillars have been discussed, our paper unifies them into a coherent framework to guide future system design.
> Building efficiently generalizing systems is a challenge the community has not yet addressed in a comprehensive, meta-level way.
>
> ### Technical depth
> In line with the ICML 2025 Position Paper CFP, we take a meta-level perspective, aiming to shift focus toward systematically addressing these six core components as guardrails and design guidelines. We invite the ML community to conceive, test, and refine systems around these dimensions. Thus, our paper broadens the conversation from "NeSy vs.\ Pure Neural" to "How do we achieve efficient generalization, and what is essential to that end?"
> We do not provide extensive algorithmic details, which would be more appropriate for a research-track paper. As noted in other responses, full implementation details exceed our scope. However, we are actively collaborating on more specialized technical work that instantiates these pillars in neuro-symbolic prototypes.
>
> ## B) Addressing Benchmarks & ARC Emphasis
>
> We acknowledge that relying heavily on ARC may appear limiting. However, ARC is one of the few benchmarks explicitly designed to minimize statistical shortcuts and emphasize *skill-acquisition efficiency*.
> By removing extraneous noise and avoiding easily exploitable patterns, ARC helps isolate how effectively a model can acquire and transfer skills from limited experience. As mentioned in Point A (with reviewer S6vw), pinpointing and measuring skill acquisition efficiency is a key research question in itself.
>
> This complements real-world deployments, where noise may obscure whether a model genuinely generalizes or simply leverages massive data to memorize patterns.
> While ARC is not perfect, it provides an environment to observe the interplay of multiple factors (architectural choices, knowledge priors, hierarchical abstractions, etc.), aligning with our six-pillar framework.
>
> ### Real-world transition
> We agree that engineering work is needed to transfer ARC insights to real-world tasks. However, ARC helps evaluate generalization in a controlled setting, offering valuable insights before tackling noisy real environments. Historically, benchmarks like chess were once considered "real-world intelligence benchmarks" but were later solved by methods lacking broader reasoning. ARC avoids this pitfall by focusing on open-ended reasoning skills.
>
> While practical deployment requires handling noise and large-scale integration, lessons from ARC can inform real-world contexts where robust, data-efficient learning is critical.
>
>
> ## D) Addressing "New Information" and Practical Experiments
>
> In a *position paper*, our core mission is to: a) Synthesize recent results (e.g., ARC, GPT~o3’s partial successes/failures in generalization) into an actionable framework. b) Propose a mindset shift for the research community, encouraging work that systematically addresses each pillar instead of pushing incremental gains in data-driven or rule-based systems alone.
>
> While we do not prescribe a single new experiment or architecture, our recommendations towards generalization (e.g., focus on skill-acquisition efficiency, exploring test-time learning, utilizing module synergies) are tangible research avenues, not a rehash of known conclusions.

---

### Official Review · Reviewer_jNXQ · 2025-03-11

**Significance:** 2
**Argument Clarity:** 2
**Rating:** 2
**Confidence:** 4

**Questions:**

-please address the concerns I raised in the weakness comments.

-the paper can benefit from more in depth study of  its core “pillars.” What are the challenges in realizing individual pillars and then orchestrating them to a general intelligence model.

-what are the actual challenges of integrating neuro-symbolic approaches in real world deployment constraints. Why such systems have not been materialized?

-Your paper emphasizes training, but how does your proposed framework support dynamic knowledge acquisition and evidence gathering during inference? Shouldn’t you envision specific mechanisms to update or augment knowledge in real time during inference?

-Given that similar hybrid approaches have been proposed previously, can you clarify what is fundamentally novel about your positioning framework? How does your proposed orchestration of modules differ from or improve upon existing neuro-symbolic systems?

-The conceptual framework is compelling, yet details on practical implementation are sparse. Could you elaborate on concrete methods, architectures, or case studies that demonstrate how these six pillars are integrated in a coherent system?

-the authors should explain more as to why the view of Chollet on the definition of intelligence is adapted. What are the alternatives and why this view is the most appropriate one..

**Discussion Potential:**

2

**Paper Summary:**

This paper posits that neuro-symbolic integration is key to achieving efficient general intelligence, critiquing both purely neural (e.g., LLM-based) and purely symbolic methods for lacking robust, scalable, and interpretable generalization. The authors adopt Chollet view that general intelligence should be measured by skill-acquisition efficiency. So they ask we should evaluate a system intelligence by examining how well the system adapts to new tasks using limited data and minimal engineering.

The paper then outlines six “pillars” of neuro-symbolic generalization: (1) multi-component synergy, (2) model specificity, (3) knowledge encoding, (4) knowledge acquisition and transfer, (5) representation learning, and (6) hierarchical abstraction, that would pave the way for general intelligence. These principles guide how to design systems that are scalable, explainable, and data-efficient. The authors acknowledge LLMs for their ability to capture patterns but criticize their opacity and reliance on massive training data. The authors also stress that purely symbolic AI methods often fail to handle real-world complexity due to lack of flexibility. Consequently, the paper contends that hybrid neuro-symbolic architectures, blending neural feature extraction with symbolic logic, are better suited for general intelligence. The authors finally make recommendations for new benchmarks, open-source frameworks, and improved integration strategies to move the field toward general intelligence.

**Position:**

Yes

**Position In Title:**

Yes

**Related Work:**

3

**Strengths And Weaknesses:**

Strengths

-The paper addresses a highly relevant and timely topic, emphasizing that integrating neural and symbolic methods is essential for achieving more efficient and robust general intelligence.

-The authors effectively outline recent developments in both neural and symbolic domains and propose a clear argument structured around six foundational “pillars” for neuro-symbolic generalization.

-There is a comprehensive review of related prior literature, highlighting the limitations of purely neural approaches (e.g., large language models) and purely symbolic frameworks. The authors convincingly demonstrate how their proposed neuro-symbolic integration approach addresses and bridges these limitations, clearly defining boundaries between neural, symbolic, and hybrid strategies.

-Building on these conceptual pillars, the authors offer actionable recommendations for guiding future research efforts.

-By emphasizing “skill-acquisition efficiency” as a critical metric, the paper directs the research community’s attention toward developing more rigorous and meaningful evaluations of general intelligence models.

Weaknesses

-While the paper advocates using datasets like ARC for benchmarking general intelligence, some real-world scenarios may not easily conform to the structured and formal nature of ARC-style datasets.

-The paper does not adequately address the role of dynamic knowledge acquisition and evidence collection during inference (rather than just during training), a critical requirement for true general intelligence models.

-The paper’s position advocating for neuro-symbolic AI integration is not entirely novel; several past works have previously highlighted similar hybrid approaches. Moreover, the current work provides limited insight into how the proposed modules should be orchestrated to clearly showcase their practical advantages, thus resembling more of a survey rather than proposing a fundamentally new research direction.

-Although the six pillars are conceptually valuable, the paper lacks sufficient detail regarding their practical implementation, methods for integrating these pillars coherently, and deeper technical elaboration.

-The discussion could also benefit significantly from explicitly highlighting how neuro-symbolic approaches facilitate a meaningful separation between reasoning and verification processes.

**Support:**

3

---

> ### Author Rebuttal · Authors · 2025-03-31
>
> Thank you for your thoughtful review of our submission. We value your insights and have addressed each of your concerns in detail below.
>
> ## A) Use of Real-World Datasets Versus ARC-Style Benchmarks
>
> We respectfully disagree that ARC-style tasks are structured and formal. If there were exploitable structural or formal invariants to these tasks, solving ARC would be much easier.
> However, we appreciate your concern that ARC might be perceived as overly structured or “toy-like.” For a detailed statement why we still think ARC is most appropriate and how it compares to "real-world" dataset, please see point B with reviewer tqCm
>
> ## B) Dynamic Knowledge Acquisition and Evidence Collection During Inference
>
> We fully agree that *dynamic* knowledge acquisition, beyond the static training phase, is essential for true general intelligence. Bulletpoint "Formation of New Concepts" (line 236) and section 4.4 (" Knowledge Acquisition, Transfer, and Combination") address how systems can acquire or refine concepts at inference time (e.g., DreamCoder's dynamic concept synthesis, references to test-time-fine-tuning and active inference).
> Given the breadth of this topic, we do not survey every approach. Nonetheless, we recognize its importance and highlight it as a key direction for systems aiming at broad generalization.
>
> ## C) Mechanisms for Real-Time Knowledge Updates and Deployment Constraints
>
> You inquired about the challenges of orchestrating neuro-symbolic systems in real-world scenarios, particularly regarding *inference-time* knowledge augmentation. We view this as an important sub-problem within our six-pillar framework (notably under "Knowledge Acquisition & Transfer" in Section 4.4).
>
> **Our paper’s scope:** As this is a *position paper*, our primary goal is to present a structured paradigm rather than a concrete technical blueprint. Indeed, the ICML 2025 Position Paper CFP explicitly encourages meta-level perspectives on the field.
>
> **Complexity in real-world deployments:** Integrating diverse complex components (be it symbolic logic with neural systems or our six pillars) requires additional engineering (e.g., domain-specific languages, interface layers). We are aware of these challenges and hope our presented findings guide researchers in systematically addressing them over time.
>
> ## D) Depth of the Six "Pillars"
>
> We appreciate your call for a more in-depth discussion of the six pillars. Our aim was to introduce these pillars clearly within the constraints of an 8-page limit and a broad, interdisciplinary audience:
> **Motivational sections (1-3):** We review definitions of intelligence, the pitfalls of purely neural or symbolic methods, and the rationale for neuro-symbolic integration.
> **Core proposal (Section 4):** We devote three pages to describing these pillars and their interplay to support efficient generalization, striking a balance between accessibility and specificity.
>
> Should space allow in the final submission, we will expand on concrete challenges and potential solutions for orchestrating these pillars (e.g., synergy between representation learning and hierarchical abstraction).
>
> ## E) Novelty & Technical Elaboration of Our Proposed Framework
>
> We kindly ask you to check point A with reviewer tqCm
>
> ## F) Differentiating Our Approach From Existing Neuro-Symbolic Systems
>
> We kindly ask you to check point A with reviewer tqCm
>
> ## G) Balancing Survey Elements and Forward-Looking Directions
>
> We acknowledge your observation that our paper includes numerous references to prior literature, which might appear “survey-like.” However, we see this as necessary grounding.
> Much research about artificially creating general intelligence is speculative. Therefore, we carefully selected many different sources to substantiate our claims. Nevertheless, we are not just stating previous developments but propose a structured perspective on how to achieve efficient generalization in the future (see point A with reviewer tqCm).
>
> ## H) Adopting Chollet’s Definition of Intelligence
>
> Finally, regarding our choice to follow Chollet’s view of intelligence as "skill-acquisition efficiency," we discuss the rationale in Sections 1.1-1.3. As you note, alternative definitions exist; however, Chollet’s relatively recent framework remains one of the few that explicitly disentangles *final performance* from the *effort or data* required to achieve it. Other definitions often conflate these factors, making it harder to isolate a system’s genuine capacity for adaptation. Given the scarcity of *actionable*, quantifiable definitions for general intelligence, we have found Chollet’s conception the most practical for the ARC benchmark and beyond.

---

### Official Review · Reviewer_wcHv · 2025-03-13

**Significance:** 2
**Argument Clarity:** 1
**Rating:** 1
**Confidence:** 4

**Questions:**

None

**Discussion Potential:**

2

**Paper Summary:**

This paper argues that general intelligence requires neurosymbolic approaches. First, the paper discusses intelligence. Then, the paper outlines how neural approaches and symbolic approaches separately aim for intelligence, and their strengths and weaknesses. Then in the second part of the paper, the paper outlines a vision for neurosymbolic AI, gives a review of existing neurosymbolic approaches, and lists some pillars of efficient neurosymbolic generalization to guide future neurosymbolic research.

## Update after rebuttal
I maintain my score. The rebuttal acknowledges differences in what I was looking for in the paper versus what they wrote, and acknowledges that the paper is more about broad conceptual pillars for guiding neurosymbolic AI research, not an attempt to prove that the path to AGI will be neurosymbolic. However, even if the authors changed the title and framing of their paper to fit this, I find the vagueness of their discussion to limit the convincingness and insightfulness of the arguments.

**Position:**

Yes

**Position In Title:**

Yes

**Related Work:**

3

**Strengths And Weaknesses:**

Overall, I don't find this to be a very convincing position paper.

The first part of the paper is supposed to be an argument for why neurosymbolic AI is needed. But comparing LLM's on one side, and then symbolic methods, and listing the downsides of each, does not come off as very convincing for why neurosymbolic will fix these issues.

I think the paper would be more convincing if it included examples of how neurosymbolic approaches have improved upon purely neural or purely symbolic in **real problems where neurosymbolic is better**. Examples of these do exist! For example, AlphaGeometry uses a symbolic enumeration of proofs in combination with a neural tactic generation model. Or lots of LLM + code models rely on "symbolic" representation of code to ensure verifiability, systematicity, etc.

Table 1 sounds like it's listing dimensions **needed** in order for models to have broad generalization. But the list sounds more like a list of desired traits, that might not be necessary. For example, a model might generalize broadly without being transparent or interpretable! It sounds like you're actually just describing overall desiderata for a useful model. So, I think this table is presented a little misleadingly, and should either be titled better, or have clearer/more justified entries.

Many of the recommendations in the paper are quite broadly stated, and as a result are a bit lacking in meaning. For example, section 4.1 says that "multi-component synergy effects" where different subcomponents strengthen each other is an important pillar for neurosymbolic systems. For example, a neural recognition system in DreamCoder strengthens search, and the DSL can be used to strengthen neural recognition. However, it's not clear what the larger takeaway or guiding message is for bigger systems. "If your system has multiple components, try to make them synergize" ? It seems like even non-neurosymbolic systems could have "multi-component synergy effects" — for example, a "self taught reasoner" LLM might improve its generative model based on its own feedback.

The rest of the pillars for neurosymbolic AI (table 3) are similarly bland and not super specific to neurosymbolic. I think they are pillars that try to capture important design decisions for neurosymbolic AI. it's an attempt to try to be more specific about directions neurosymbolic AI should go, but to me it just exposes a lack of detail in the paper's vision for neurosymbolic AI. Because it's so broad, it suggests that there isn't a clear roadmap for how to improve neurosymbolic AI, other than taking existing approaches' strengths and looking more into those strengths.

**Support:**

2

---

> ### Author Rebuttal · Authors · 2025-03-31
>
> Thank you for your time and for reviewing our paper. We appreciate your feedback and suggestions, which have helped us refine and clarify our position. Below, we address your main concerns in detail.
>
> ## A) Convincing Arguments for Neuro-Symbolic Approaches
>
> As per the official guidelines (ICML 2025 Position Paper CFP), we devoted Section 2 to discussing viable alternatives to our proposed multi-component approach. Specifically, we contrast: a) **Purely neural methods,** which excel at pattern recognition and scalability but may struggle with certain forms of symbolic manipulation or explicit reasoning. b) **Purely symbolic methods,** which are well-suited to interpretability, rule-based reasoning, and combinatorial search but often lack the flexibility and inductive bias that data-driven neural models provide.
>
> To repeat from Section 3, "Neuro-Symbolic Approaches": Neuro-symbolic designs aim to *combine* the strengths of both paradigms while mitigating their individual weaknesses. When integrated carefully, the synergy that emerges enables a system to *generalize across a broader range of tasks* than purely neural or purely symbolic methods alone.
> We do provide evidence [1] and references demonstrating how merging these paradigms fosters more efficient generalization.
>
> Additionally, we kindly ask you to check out discussion point A) with reviewer tqCm where we state how our view goes beyond the classical neuro-symbolic argumentation.
>
> Through your review, we have seen that in the final version, we need to clarify that this paper does not aim to prove the universal superiority of neuro-symbolic AI.
>
> ## B) Examples of Neuro-Symbolic Approaches in Practice
>
> We appreciate your suggestion to include more concrete examples of successful neuro-symbolic systems (e.g., AlphaGeometry, or LLM+code models that leverage symbolic representations for verifiability). Our current manuscript focuses primarily on *conceptual pillars* of efficient generalization rather than performing an exhaustive survey of existing neuro-symbolic solutions. That said, we do reference specific works in Table 2 and throughout Section 3 that illustrate how combining symbolic and neural paradigms leads to systematic improvements in tasks such as the ARC challenge.
>
> In the final version of our submission, we will **reiterate that our conceptual framework** is intended to provide a *design paradigm* for generalizable AI, one that must be instantiated by these real-world systems.
>
> ## C) The Scope of Our Position Paper
>
> You raised concerns about the breadth of our recommendations and how they might apply to real-world problems. As detailed in point A with reviewer tqCm, our paper aims to *articulate a conceptual framework* of six key pillars that, in our view, are necessary to achieve robust skill-acquisition and generalization. We focus on core *design dimensions* - from knowledge encoding and modular design to representational hierarchies and abstractions - that we believe underlie successful systems.
>
> While some points could appear “high-level” or “bland,” our motivation is to guide researchers toward systematically integrating multiple components and ensuring that these components *synergize* to achieve effective generalization. Indeed, this is not exclusive to neuro-symbolic AI; our suggested concept is orthogonal to classical neuro-symbolic design paradigms. We merely argue that neural-symbolic approaches are particularly promising when aiming for our multi-component synergies.
>
> We will adapt the introductory formulations in section 4 respectively to make this distinction in scopes more obvious.
>
> ## D) Table 1: Dimensions for Broad Generalization
>
> We acknowledge your observation that certain system properties (e.g., transparency or interpretability) are not strictly required for a model to generalize well on some tasks. However, as discussed in Section 1.2, *our focus extends beyond mere final performance*. We aim to offer *design guidance* for more robust, reliable, and maintainable systems. Hence, the dimensions in Table 1 (titled "Dimensions of Designing Models with Broad Generalization") serve as *guardrails* to help researchers reason about the developmental process, rather than as an absolute “checklist” for an end system.
>
> To avoid confusion, we will clarify in the text that these are *suggested design principles* rather than mandatory criteria. Also, we will **elaborate on each dimension’s role** in enabling generalization and how it ties into a broader system design perspective.
>
> ## E) Specificity of the Six Pillars
>
> Please check discussion point A) with reviewer tqCm for the scope and novelty of the pillars. We refer to point B) with reviewer S6vw for a detailed description of the meaning of the pillars.
>
> [1] Bober-Irizar, M. and Banerjee, S. Neural networks for abstraction and reasoning.  November 2024.

---

### Official Review · Reviewer_S6vw · 2025-03-14

**Significance:** 3
**Argument Clarity:** 2
**Rating:** 4
**Confidence:** 3

**Questions:**

Questions,  comments and suggestions.

- line 49, section 1.1. Regarding this passage below, I don't think
minimal prior knowledge is an issue: an intelligence that makes **effective** use
of its knowledge, whether broad/vast or small, seems fine.  The
issue to me is just determining what efficiency or effectively really means (in
acquiring and use of knowledge)!  (similarly for better
understanding/quantifying robustness, safety/reliability, interpretability, and so on)

The passage: "This perspective shifts attention from raw performance on a single
task to the ability to learn new tasks under constraints – such as
limited data, novel transformations, or minimal prior
knowledge. Indeed, skill-acquisition efficiency is at the heart of
what sets “general” intelligence apart from specialized or
over-engineered solutions (Bober-Irizar & Banerjee, 2024)."

- In Table 3, key pillars: representation (4.5) and knowledge encoding
(4.3) seem to be the same ideas or issues ... and very closely tied to
abstraction and hierarchies (abstraction and/or hierarchies are sub
pillars, of representation and/or knowledge encoding ...  the authors
may have a different idea in mind of knowledge encoding, such as 'factual' or
declarative knowledge, vs 'skill knowledge' or semantic memory, etc,
such as how to talk or ride a bike.. or what 'representation' means..  ) ...

- The term 'deverloper-aware' is not really defined/explained in this paper (a citation to Cholet 2019 is
provided).  I think I figured it out, and it is probably better to use 'deverloper-free' or 'independent'/autonomous...
(in any case, a short additional explanation would be useful)

**Discussion Potential:**

3

**Paper Summary:**

The authors argue that currenly LLM models and techniques, while
broadly applicable, still have limitations, in particular in reasoning
and efficient skill acquisition. On the other hand, mixed
neuro-symbolic approaches offer more robust generalization across
diverse tasks.  The authors state that additional factors – such as
modular transparency, and more flexible representations are necessary
to further advance generalization capability.

The authors advocate focusing on the ~6 pillars that they propose and further working on tasks such ARC-AGI benchmarks.

"## update after rebuttal"

I read the authors' clarifications and responses to the other reviews.  I believe the authors can improve the paper clarity (the points they want to make), and do have a useful position for the ICML position.  I have increased my position to borderline-accept to accept.

**Position:**

Yes

**Position In Title:**

Yes

**Related Work:**

3

**Strengths And Weaknesses:**

Strengths: describing a number of efforts and how they overcome the challenges in the skill/reasoning benchmarks.

Weaknesses: the proposed capabilities or directions to be investigated require more clarity (see questions).

**Support:**

2

---

> ### Author Rebuttal · Authors · 2025-03-31
>
> Thank you for the in-depth review and the valuable points you raised! Below, we address them in detail.
>
> ## A) Prior Knowledge & Skill-Acquisition Efficiency
>
> An agent that extracts greater competence (skills, insights, etc.) from *identical* training conditions is inherently more efficient at acquiring skills. In evolution, this efficiency holds an inherent advantage. Most modern intelligence tests are based on the related concepts of fluid and crystallized intelligence.
> Figure 3 from [1] visualizes this "information conversion ratio" from situational to operational space nicely.
> Consequently, skill-acquisition efficiency, i.e., the amount of competence gained from a fixed amount of data or experience, should be evaluated independently of a system’s final performance. The prior knowledge of a system matters as it gives a heads-start on performance.
> Traditional benchmarks often conflate performance with the data or developer-engineered interventions needed to achieve it. Here is where ARC comes in; it tries to isolate how a system’s **learning efficiency** emerges (and how it might be improved). All other factors (e.g., curriculum size, development efforts (e.g., inductive bias), training time & strategy, intrinsic task difficulty) should be controlled for to isolate the system's skill-acquisition efficiency.
>
> We emphasize that understanding and quantifying skill-acquisition efficiency is an open research question. We try to raise awareness of its relevance for genuinely general AI.
>
> We hope that the above explanations provide sufficient context. We will clarify the respective sections in our paper respectively for the final version.
>
> ## B) Conceptual Distinctions Among the Pillars
>
> The pillars in Table 3 are a compressed summary of the detailed descriptions in Sections 4.1-4.6. We identified these six pillars that frequently appear in successful generalization approaches.
>
> ### How the pillars differ:
>
> Knowledge Encoding (4.3) focuses on how human competence is “injected” into a system (e.g., via curated datasets, architectural biases, or manual hyperparameter tuning). It addresses what competence is *pre-loaded* into a model before training.
>
> Representations (4.5) are the internal *model of reality*, they define the complexity and structure of the system's "world model", whether continuous embeddings or discrete symbols. They can be engineered (e.g., ontological graphs) or emerge purely data-driven (e.g., embedding spaces). Representation spaces influence how the environment is perceived and processed, ultimately shaping what abstractions and inferences are possible. While subject to knowledge encoding and abstractions, they are a distinct aspect/pillar of a system.
>
> Abstractions and Hierarchies (4.6) define the ability to transform raw input into progressively more conceptual representations by discarding irrelevant details. A CNN, for example, hierarchically abstracts raw pixels into high-level semantic features. This allows to process initially different things as similar at a higher level, which is central for generalization. The system recognizes structurally similar scenarios and transfers learned skills more readily. This flexibility is a distinct phenomenon from merely engineering knowledge (4.3) or having a fitting internal representation (4.6)
>
> Although these three pillars (and the others in our paper) are inherently interrelated, each highlights a distinct mechanism supporting or constraining a system’s competence to generalize over novel tasks.
>
> ### How the pillars interrelate:
>
> The first 3 paragraphs of Section 5 describe how the pillars interconnect and their interdependence. We write, that their interplay is precisely what matters for generalization (and intelligence). A system that maximizes one pillar and minimize the others would be ineffective. Useful, clever, and flexible systems are well-developed in all of these pillars. We observed approaches to implicitly optimize for these pillars. Our paper makes this focus explicit.
>
> We hope that the above explanations provide clarity.
> We acknowledge that the current descriptions in the paper could highlight the differences and interrelations more clearly. We will revise these sections for the final version of the paper!
>
> ## C) The Term “Developer-Aware”
> As you pointed out, "developer-aware" is defined in our reference [3, p.10]: "Developer-aware generalization: this is the ability of a system [...], to handle situations that neither the system nor the developer of the system has encountered before." Further, "Note that "developer-aware generalization" accounts for any prior knowledge that the developer of the system has injected into it."
>
> We will include a short definition in our final submission for clarity, especially for readers unfamiliar with Chollet’s work.
>
> We hope to have addressed all your concerns satisfactorily and would be thankful for a re-evaluation of our submission.
>
> [1] Chollet, F. On the Measure of Intelligence, November 2019

---

### Decision · Program_Chairs · 2025-04-27

**Decision:**

Reject

**Comment:**

While the paper advocates using datasets like ARC for benchmarking general intelligence, some real-world scenarios may not easily conform to the structured and formal nature of ARC-style datasets. The paper would be more convincing if it can include examples of how neuro-symbolic approaches have improved upon purely neural or purely symbolic in real-world problems where neuro-symbolic is better.
In addition, the paper does not adequately address the role of dynamic knowledge acquisition and evidence collection during inference (rather than just during training), a critical requirement for true general intelligence models.